# Principles and Applications of Seismic Monitoring Based on Submarine Optical Cable

**DOI:** 10.3390/s23125600

**Published:** 2023-06-15

**Authors:** Junzhe Yu, Pengbai Xu, Zhangjun Yu, Kunhua Wen, Jun Yang, Yuncai Wang, Yuwen Qin

**Affiliations:** 1Provincial Key Laboratory of Photonics Information Technology, School of Physics and Optoelectronic Engineering, Guangdong University of Technology, Guangzhou 510006, China; 2Provincial Key Laboratory of Photonics Information Technology, School of Information Engineering, Guangdong University of Technology, Guangzhou 510006, China

**Keywords:** fiber-optic sensors, seismic monitoring, submarine cables, distributed acoustic sensing

## Abstract

Submarine optical cables, utilized as fiber-optic sensors for seismic monitoring, are gaining increasing interest because of their advantages of extending the detection coverage, improving the detection quality, and enhancing long-term stability. The fiber-optic seismic monitoring sensors are mainly composed of the optical interferometer, fiber Bragg grating, optical polarimeter, and distributed acoustic sensing, respectively. This paper reviews the principles of the four optical seismic sensors, as well as their applications of submarine seismology over submarine optical cables. The advantages and disadvantages are discussed, and the current technical requirements are concluded, respectively. This review can provide a reference for studying submarine cable-based seismic monitoring.

## 1. Introduction

Oceans are of great importance for the living beings on the Earth as they provide abundant resources and space. The sparsity of valid observation methods leads to a biased recognition of oceans, limiting the availability of information on subsea geohazards [1], Earth’s interior structure [2], underwater seismology [3], and oceanography [4]. Hence, exploiting marine observation techniques is necessary for long-term sustainable development.

Many countries [5,6,7,8] around the world have already carried out research on submarine seismic monitoring. They put monitoring devices, such as strong motion seismometers [9], broadband seismometers [10], and manometers [11], into several hundred meters below the sea to detect and analyze earthquake events. Most of the currently used submarine seismic sensors are with electronic structures that require a power supply, which makes the seismometers susceptible to electromagnetic interference, seawater corrosion, and deep-sea high pressure. Moreover, if the seismic waves are generated from tens to thousands of meters away, single-point seismometers face the challenge of inadequate information obtainment. Although the integrated network of multiple seismometers can collect extra data, the asynchronous data obtained from each seismometer restricts the real-time access ability. 

The fiber-optic sensor is a kind of all-optical sensing means without any electronic devices. It is immune to electromagnetic interference and shows better anti-seawater-corrosion ability. Benefiting from its low transmission loss and small scale, fiber-optic sensors can be integrated into a wide sensing network up to hundreds of kilometers long. Distributed optical fiber sensors, which prosperously developed in the recent thirty years, can simultaneously measure a million sensing points over a hundred kilometers, showing high measuring density and excellent synchronization of data acquisition. Compared with electronic seismometers, the fiber-optic sensor can overcome the above inherent limitations, which shows potential applications for seismic monitoring.

The submarine optical cable built by telecom carriers is the backbone of international and intercontinental telecommunication. There are 552 submarine optical cables worldwide with a total length of over 1.4 million kilometers by early 2023 [12]. This abundant cable network offers a significant basis for seismic monitoring [13]. Recently, the SMART Cables system (Science Monitoring and Reliable Telecommunications), which deploys sensors on telecommunications cables, has been proposed to obtain extensive, longitudinal, and real-time observation data for the early warning of earthquakes. The International Telecommunication Union (ITU), the World Meteorological Organization (WMO), and the Intergovernmental Oceanographic Commission of the United Nations Educational, Scientific and Cultural Organization (UNESCO/IOC) established the Joint Task Force to promote the development of the system. Several simulations have demonstrated its feasibility and further practical pilots are expected to be conducted before 2025 [4,14]. Meanwhile, there are several research studies that use submarine optical cables for seismic monitoring [15,16,17,18], which demonstrates that the implementation of submarine optical cables could extend the detection coverage, improve the detection quality, and enhance long-term stability for earthquake monitoring. 

In this paper, we discuss the potential for submarine earthquake monitoring that adopts fiber-optic sensing technology based on submarine cables. The main principles of optical seismometers based on submarine cables include optical interferometers, fiber Bragg grating (FBG), optical polarimeter, and distributed acoustic sensing (DAS). For these techniques, the submarine optical cables could not only be utilized as the transmission media, but also serve as the sensing element. The optical interferometers and fiber Bragg grating (FBG) mainly adopt submarine cables for data transmitting; meanwhile, the optical polarimeter and DAS system utilize submarine cables to perceive external information. Both operation principles and practical applications of these optical seismometers are introduced. We analyze the advantages and disadvantages of each fiber-optic technique. Last, we discuss the features of different fiber-optic sensing techniques and provide an outlook of their future development. 

This article is organized as follows: in Section 2, we introduce the operation principles of different optical seismometers based on the optical interferometer, FBG, optical polarimeter, and DAS. In Section 3, we outline the corresponding submarine seismic monitoring applications of these fiber-optic sensors and discuss their advantages and disadvantages. In Section 4, we summarize the features of these submarine cable-based fiber-optic sensing techniques and look forward to their further development. 

## 2. Fiber-Optic Sensing Techniques for Seismic Monitoring

### 2.1. Principle of the Optical Interferometer-Based Seismic Monitoring

The principle of the optical interferometer is that the external vibration can be restored by the variation in the optical phase difference of the interferometry signal. The main structures are Mach–Zehnder interferometers (MZI), Michelson interferometers (MI), and Fabry–Perot interferometers (FPI).

The structure of MZI is shown in Figure 1a. The light emitted from the laser is divided into two beams, serving as the reference light and the sensing light, respectively. After passing through two arms, the beams are recombined and interfered with each other at the coupler and detected by two detectors individually. On the upper sensing arm, the optical fiber is multi-wound on a mass block tightly [19]. If a vibration occurs, the mass block would induce a stretch on the coiling fiber and then an optical phase shift that carries the vibration information is produced. This vibration-induced phase shift is transferred to amplitude variation which is used to retrieve external perturbances. The sensing technique of MI is similar to that of MZI. As shown in Figure 1b, the main difference is the reflectors, which are used to reflect beams in the two arms. The disturbance acting on the sensing arm is detected as phase changes, which is then used for quantitative measurement. The FPI sensor uses two parallel reflectors with a certain distance to form a resonant cavity inside the sensing fiber. As shown in Figure 1c, both the transmitted and reflected beams interfering with each other in the cavity multiply, leading to precise fringes with high spectral resolution. The phase difference of the FPI is given as,
(1)$$\delta_{FPI}=\frac{4\pi }{\lambda }nL$$
where λ is the wavelength of the incident light, n is the refractive index of the cavity, and L is the length of the cavity. When the perturbation is introduced to the cavity, the optical phase difference is influenced by the changes in the cavity length and refractive index, resulting in a shift in wavelength. By measuring the shift of the wavelength spectrum, the applied strain can be quantitatively calculated [20]. The optical interferometer applied for submarine seismic monitoring should have a long sensing arm with a high sensitivity to realize micro-seismic detection. 

### 2.2. Principle of the FBG-Based Seismic Monitoring

FBG can be deemed as an optical reflective mirror, where a periodic variation of the refractive index is formed in the fiber core [21]. Figure 2 shows the structure of FBG. The light going through the gratings can be maximum reflected if the Bragg condition is satisfied. The central wavelength of reflected light is determined by,
(2)$$\lambda_{B}=2n_{\mathrm{eff}}\Lambda$$
where $n_{\mathrm{eff}}$ is the effective core refractive index, Λ is the grating pitch, and $\lambda_{B}$ is the Bragg wavelength. The applied perturbance can be quantitatively demodulated through the measurement of the wavelength shift. The FBG applied for submarine seismic monitoring should have a high measuring precision and a broad response bandwidth to detect seismic signals.

### 2.3. Principle of the Optical Polarimeter-Based Seismic Monitoring

Polarimeter sensing is achieved by establishing a relationship between the polarization state of the transmitted light and the external perturbances. As shown in Figure 3, the state of polarization (SOP) in the fiber can be represented by Stokes vectors, which are the unit vector in the Poincaré sphere in a three-dimensional Stokes space. The shift of birefringence, which carries perturbance information, can be obtained from the deviation of output SOP in the Poincaré sphere [22],
(3)$$\langle \Delta s^{2}\rangle =\frac{\pi }{4}{\left(\frac{2\pi c}{\lambda }\right)}^{2}\int_{0}^{z}\kappa^{2}\frac{\langle \Delta \beta^{2}\rangle }{\langle \beta^{2}\rangle }dz^{\prime }$$
where $\Delta s^{2}$ represents Stokes vector displacement, which can be calculated from the spectrogram of polarization states. κ is the polarization mode dispersion coefficient of the cable. $\Delta \beta^{2}$ means variations of birefringent disturbance, which is mainly affected by a change in the effective refractive indices of the two eigenpolarizations and the stretching of the fiber length. The change of the effective refractive indices is almost induced by external strain ε in the standard single-mode fiber due to its circular symmetry. Therefore, birefringent disturbance $\Delta \beta^{2}$ can be rewritten as $\varepsilon^{2}\beta^{2}$ and Equation (3) can be simplified as a quantitative relationship between SOP and strain [23]. Then, the external strain variation can be retrieved by Stokes vector displacement. The optical polarimeter applied for submarine seismic monitoring should have a high sensitivity and long-term operation stability to monitor seismic signals.

### 2.4. Principle of the DAS-Based Seismic Monitoring

DAS is a distributed optical fiber sensing technology that uses the backward Rayleigh scattering in the optical fibers. It can retrieve the external vibration along the optical fiber by demodulating the phase or intensity information of Rayleigh backscattered light. Two main structures can be categorized into coherent detection-based DAS and direct detection-based DAS, respectively. The structure of coherent detection DAS is shown in Figure 4 with dashed lines. The light emitted from the laser that has narrow linewidth is separated into two beams by a coupler. One beam serves as a local reference light, and the other beam is the probe light. The probe light is then modulated as the probe pulse. The probe pulse is launched into the fiber under test (FUT) through an optical circulator after amplification. The resulting backscattering signal is then interfered with the local reference light and recorded by the photodetector. If a vibration occurred, the refractive index of FUT would change, leading to an optical phase shift. The variations of phase are extracted from the beat signal for vibration demodulation [24]. The structure of direct detection DAS is shown in Figure 4 without dashed lines; it utilizes an external cavity laser driven by an arbitrary function generator to generate chirped light as the probe signal without a local reference light [25]. The rest devices work similarly to coherent detection-based DAS. Through the frequency-to-time mapping characteristic of chirped pulses, it can convert perturbation-induced spectral shift into a local temporal delay in the Rayleigh backscattering trace [26]. The external perturbances can be quantitatively demodulated with the cross-correlation algorithm among the collected Rayleigh backscattered time domain traces. The coherent detection-based DAS applied for submarine seismic monitoring should have a good low-frequency response to detect microseismic signals. Meanwhile, the direct detection-based DAS should have a long sensing distance to monitor teleseism signals.

## 3. Applications of Seismic Monitoring Based on Submarine Cable

### 3.1. Seismic Monitoring Based on Optical Interferometer

The Japan Agency for Marine-earth Science and Technology first developed an MI-based fiber-optic accelerometer to measure a submarine earthquake in 2002. Its measuring sensitivity of the micro-seismic signal is 30 ng/Hz^1/2^, and its frequency response is in the range from 0.3 to 200 Hz [27]. The University of California exploited a submarine optical strainer based on a pair of MI to realize long-term seismic monitoring in 2018. The two sensing fibers in MI have different thermal responses to resolve temperature fluctuation [28]. Harbin Engineering University analyzed the noise distribution of interferometry seismometers to provide the basis for low-frequency noise suppression in 2021 [29]. The Institute of Semiconductors, Chinese Academy of Sciences designed a low-frequency FPI seismometer based on global noise level analysis to monitor low-frequency seismic in 2021. Its frequency response is in the range from 0.01 to 50 Hz and its noise level is 6.74 ng/Hz^1/2^ within 1 to 50 Hz in the laboratory [30]. Although interferometry seismometers have realized long-term stable seismic detection, it shows the difficulty in monitoring very low-frequency earthquake waves due to the laser phase noise and submarine environmental noise.

According to the above work, one factor affecting submarine seismic monitoring is the submarine environmental noise. In 2018, the National Physical Laboratory proposed Ultra Stable Laser Interferometry (USLI) [15] based on the MI and the submarine optical cable. They utilized the USLI system to detect submarine seismic signals and compared them with the signals recorded by seismic stations on land. As shown in Figure 5a, an ultra-low-expansion (ULE) glass Fabry–Perot cavity in the submarine link is utilized to lock the laser phase. The ultra-stable laser pulses are emitted into the submarine link through one end of a standard fiber pair. The transmitted light is returned to the incident end by another fiber of the fiber pair. The external perturbations caused by submarine earthquakes are recovered by measuring the phase difference of the returned optical signals. Figure 5b shows the comparison between the phase changes detected by the USLI system and the displacement recorded by seismometers located along the cable. The arrival time of the detected seismic corresponded to the smallest distance between the epicenter and the fiber link. The waveform of the earthquake signals detected by the USLI system is in agreement with the seismometers. The USLI system utilizes the ULE Fabry–Perot cavity [31] to lock laser frequency and extend data coverage to 100 km, offering teleseism signals with high fidelity and strong response, whereas the seismic information detected from a single submarine link is inadequate to locate the earthquake source place. 

In order to triangulate the location of earthquake sources with multiple cables, Marra et al. utilized several 46-km-long submarine cable sections, which were partitioned by repeaters, to locate the earthquake source. As shown in Figure 5c, the repeaters contain a high-loss loop back (HLLB) path to partially reflect transmitted light to the incident end [18]. The returned signals are separated through the reflection of FBG in each repeater. The round-trip cumulative optical phase difference between the incident end and the selected repeater is extracted to realize vibration measurement. The position of the earthquake source can be located through multiple monitoring sections along the submarine cable. Figure 5d displays the frequency fluctuations of the optical signals, which are induced by earthquake events over the cable spans. The time delay between the profiles of the P waves can determine the source place of the seismic event. 

The University of Sannio developed an MI-based fiber-optic hydrophone for earthquake monitoring in 2023 [32]. The average noise floor of the system is about 100 μPa/√Hz in 1–80 Hz. In the field experiments, several earthquake events were detected, and the results were compared with a reference hydrophone sensor. Both hydrophones accurately sensed and recorded the seismic waves and the correlation coefficients between the recorded trials are higher than 85%.

The application of interferometers in seismic monitoring could conduct micro-seismic and teleseism real-time monitoring from remote offshore areas. When adopting the frequency locking technique, interferometers are almost independent of the laser frequency fluctuation and environmental noise. Then, the optical interferometers could realize very low-frequency and long-term earthquake detection in a quiet submarine environment. Meanwhile, optical interferometers could also realize the localization of the submarine earthquake through the submarine cable-based observation array. 

### 3.2. Seismic Monitoring Based on FBG

In 2007, the NTT Corporation in Japan developed an FBG-based submarine earthquake early warning system [33], which was based on several hundred kilometers-long marine cables. The frequency response ranged from 5 to 50 Hz, and the working depth was 4 km. The deployment of the system can realize early detection of near-field earthquakes. In this system, the FBGs serve as the sensing cavities and the shift of the wavelength reflects the action of seismic waves. In 2008, the Optoplan AS Corporation in Norway developed a fiber-optic multi-component submarine seismic monitoring system for permanent installations. The system, based on FBG, contained three-component accelerometers with a frequency response ranging from 3 Hz to 250 Hz. The system exhibited excellent seismic response via the active source tests, which could monitor submarine reservoir resources [34]. In this system, the FBGs serve as the networking equipment to realize complex wavelength multiplexing, extending sensing length. In 2015, the Institute of Semiconductors, Chinese Academy of Sciences, proposed an FBG-based three-component fiber submarine seismometer and carried out a series of active seismic source (air gun excitation) detection experiments in Dayindian Lake of Yunnan Province [35]. In this system, the FBGs serve as reflectors, while the sensing element is the fiber laser. In the field experiment, both the optical seismometer and the electrical seismometer were used to record the direct waves and surface waves excited by the air gun, respectively. The time domain waveforms of them have a high correlation coefficient up to 0.9897, as shown in Figure 6. This evidences the feasibility of the FBG-based submarine seismometer for seafloor application [36,37]. 

The FBG sensor shows long-distance sensing ability and up to nano-strain sensitivity. However, it is difficult to measure low-frequency seismic waves below 1 Hz due to the phase noise and frequency jitter of the laser. Although He et al. utilized a reference FBG resonator [38,39] and narrow linewidth tunable laser to compensate for laser low-frequency frequency noise and achieve the low-frequency measurement of 0.01 Hz, the practical applications for submarine earthquake detection of these low-frequency broadband FBGs have not been reported yet. 

### 3.3. Seismic Monitoring Based on Optical Polarimeter

In 2020, Zhan et al. utilized the state of polarization of transmitted light signals of a transcontinental seafloor cable to monitor seismic waves. As shown in Figure 7a, a 10,000-km-long submarine cable was employed to connect Los Angeles and Valparaiso, along the eastern edge of the Pacific Ocean. The spectrogram of Stokes parameters was analyzed to discriminate different signal components. Strong perturbances were found in the frequency band of 0.1 to 0.5 Hz, which correlated well with the expected seismic waves. This proves that the SOP variation of the submarine cable could be applied for seismic detection [40]. The comparison between earthquake signals detected by the optical polarimeter and the nearby seismic station is shown in Figure 7b [16]. It can be found that the detected seismic signals last more than 20 min, which is longer than the signals detected by the seismic station. 

The major challenge of the optical polarimeter is that an earthquake event is difficult to localize through a single submarine cable. Differential travel times of at least three cables, which have different geometry and accurate time synchrony, could be utilized to triangulate the earthquake source. In 2022, Zhan et al. activated a polarization sensing network with six long-distance transoceanic submarine cables. They utilized two of these cables, PAC-1 and PAC-2, to test the earthquake localization. As shown in Figure 7c, the SOP signals indicate that both cables detected the earthquake event. The differential travel time of the profiles between PAC-1 and PAC-2 is measured. The differential time gives constraints on the distance from the source place between the two cables at the detection point and the rough position of the earthquake source place could be obtained [23]. 

The established polarimeter system, based on the submarine communication transoceanic cable over 10,000 km, has brought a large increase in data coverage for large portions of the seafloor. The duration of the signal detected by the system is longer compared to the seismic station because it reflected the integrated shaking along the transoceanic cable. Benefiting from broader data coverage and longer-duration response, the system is expected to provide faster and more reliable earthquake warnings. Again, the background environmental noise would accumulate over the entire cable length, which limits the detection of smaller-sized earthquakes for the polarimeter system. Moreover, the fiber birefringence, which is sensitive to environmental fluctuations, would influence the precision of earthquake detection.

### 3.4. Seismic Monitoring Based on DAS

The aforementioned techniques have improved the long-term stability and large coverage of the submarine earthquake monitoring technology. The recently proposed DAS technique brings unprecedented high-density and full-continuous monitoring into submarine seismic detection. In 2009, Farhadiroushan et al. developed the intelligent distributed acoustic sensor (iDAS) interrogator based on coherent detection DAS [41], providing a new distributed seismic sensing tool. The coherent detection DAS technique employs a coherent reception scheme with a local oscillator gain in the Rayleigh backscattered signal, which brings the advantage of a high signal-to-noise ratio (SNR) [42]. However, due to coherent detection-induced interference fading, the SNR of random scattering points considerably deteriorates. It causes serious distortion of perturbance information in differential phase demodulation and limits the effectiveness of sensing [43]. Moreover, it also encounters the challenge of measuring low-frequency signals due to the accumulated phase noise between the probe signal and the local oscillator [44]. In 2019, researchers at the University of California, Berkeley, connected the iDAS interrogator with the submarine cable of the Monterey Bay Acceleration Research System. They selected the first 20 km dark fibers inside the cable offshore as the sensing arrays and carried out a series of marine geophysical experiments. The raw strain data are processed by frequency-wavenumber filtering to remove dominant ocean environmental noise and optical noise. As shown in Figure 8a,b, the propagating apparent velocities of strong coherent seismic energy, which presents the velocity of a wavefront in a certain direction, are significantly slowed between 15.1 and 16.3 km. This region corresponds to the mapped fault zones. The slowdown of apparent velocities is because the seafloor faults could act as point scatters to convert body waves into slower surface waves. The potential submarine fault zone could be inferred based on this phenomenon.

In 2021, Cheng et al. [45] utilized ambient noise autocorrelation to further characterize the seafloor structure. The ambient noise autocorrelation is applied to extract coherent signals from the raw data recorded by DAS [46]. Apparent lateral variations along the 20-km-long submarine cable with a high resolution of 20 m could be found in Figure 8c. The dashed line on the profile represented boundaries, where surface wave scattering happens and causes a decrease in the apparent velocities. A running window filter is applied to enhance these inconspicuous low velocities zones. The low velocities zones are generated mainly between 5.5 and 9.5 km at discontinuity boundaries. Figure 8d shows the resulting submarine structure image. Several low-velocity zones are found to exist at the 750 m/s velocity of the S-waves (V_s_) counter, including a deep-seated anomaly zone. The zone near 9.5 km, with sharp lateral boundaries and 400 m vertical depth, could be assumed as an unmapped fault zone. The results of ambient noise autocorrelation provide improved constraints on shallow submarine structure features, which contributes to the further characterization of the seafloor structure. 

On the other hand, direct detection DAS has a more concentrated noise distribution than coherent detection DAS because it is almost immune to phase demodulation errors caused by interference fading. Thus, the SNR of direct detection DAS is more stable along the fiber [47]. Compared with the coherent detection scheme, the laser phase noise of direct detection DAS only accumulates within the pulse width instead of the whole sensing fiber, accumulating much less phase noise than that of coherent detection DAS [48], which is preferred for seismic monitoring compared with coherent detection DAS. The high-fidelity distributed acoustic sensor (HDAS) interrogator based on direct detection DAS was released by Aragon Photonics in 2019. Several field experiments with HDAS for submarine seismology have been carried out in Europe. Williams et al. utilized HDAS to detect seismic waves which occupied a 42-km-long dark fiber inside existing submarine cables [49]. Figure 9a displays the comparison between earthquake signals detected by HDAS and the nearby broadband seismometer. The recovered P-phases and S-phases of DAS are similar with those recorded by the broadband seismometer, whereas the P-phases from DAS show low fidelity because the background noise is strong for the seismic waves.

The previous study verifies that the DAS system could detect submarine earthquake signals, while the quality of the recorded signals is dependent on the background noise, such as cable coupling levels and instrumental sensitivity. To explore the features of the DAS signals, Ugalde et al. studied and evaluated the noise level of the seismic signal obtained on a 60 km submarine optical cable. They found that channels with high SNR also experienced a strong fluctuation even if the instrumental noise was low along the cable. This was because the irregular water depth of the submarine cable and the different local coupling levels influence the quality of the detected earthquake signals [50]. 

To conduct a more reliable earthquake detection, Ugalde et al. adopted to observe hydroacoustic T waves propagating in the deep ocean to monitor submarine seismic events. T waves are earthquake-excited acoustic waves, which propagate in the slowest deep ocean channel with low energy attenuation. Therefore, T waves could transmit small-sized teleseism signals with high fidelity. 

Figure 9b shows the raw strain data of the hydroacoustic waves for one single channel of the array. The T waves, which have greater strain values than those of P, S, and surface waves, are recorded by DAS with a long duration. Strong acoustic energy is observed in the range from 1 to 10 Hz. The detection of the deep ocean T waves provides an effective approach to monitoring small earthquake signals in remote oceanic regions. 

In addition, the DAS system integrated with submarine optical cables had excellent sensitivity to monitor regional micro-seismic events. This capability was first demonstrated by Sladen et al. [51]. They successfully recorded the microseismic event of M 1.9 at the distance of 100 km through the submarine DAS arrays. The detected seismic phases were similar with that obtained by a broadband seismic station. Recently, a smaller magnitude microseismic event of Mw 1.0 was recorded by photonic integrated sensing and communication system [52]. In this system, the existing monitoring channels of the fiber-optic communication system were used as the sensing cable of the DAS to realize submarine seismic monitoring. In the field experiments, more than four microseism events are observed, which are consistent with the records from the seismic station. The magnitudes of these microseisms are below Mw 2, and the lowest one is Mw 1.1. In contrast, some observed offshore microseisms are not recorded by conventional seismic stations due to the limited station deployment. The applications of DAS for seismic monitoring based on submarine cables prove that DAS can serve as an earthquake observation array. This is because it contains thousands of sensing points that can provide massive seismic data to conduct real-time monitoring of micro-seismic and teleseism. The DAS array can also perform as an imaging tool to detect unknown fault zones and characterize submarine structures at a high resolution. However, restricted by the power-limited backscattered signals and the fiber nonlinear effects, the sensing distance of DAS for earthquake detection is generally ranging from tens of kilometers to hundred kilometers (without repeater amplification), which prevents its further seismology application.

Moreover, the DAS system can also serve as hydrophones for earthquake detection [53]. In 2021, Shanghai Institute of Optics and Fine Mechanics, Chinese Academy of Sciences, developed an underwater DAS array to realize the orientation of underwater acoustic signal source [54]. In the same year, Matsumoto et al. utilized the submarine DAS system to conduct underwater seismic acoustic waves monitoring around Shikoku Island in Japan. In the air-gun shot field experiments, the hydroacoustic signals recorded by DAS have a lower frequency response to 0.1 Hz, while the submarine self-noise is higher in the frequency range above 5 Hz compared with the nearby hydrophones [55]. It has been shown that seismic signals induced by a regional earthquake can be identified in the DAS measurement.

Seismic waves are broadband signals with frequencies ranging from 10^−4^ to 100 Hz, while frequencies between 0.01 and 10 Hz are typically used in earthquake detection [56]. Excellent low-frequency response is the premise to detect entire earthquake signals. Several seismically active areas are hundreds of kilometers away from the island or mainland. Seismic events occurring in these areas would become weak by the time they reach the seismometers, resulting in distortion of the detected signal. Broad coverage is necessary to detect high-fidelity earthquake signals. Therefore, the DAS systems designed for submarine earthquake detection should possess the ability to realize very-low-frequency detection over a long distance. Coherent detection-based DAS is able to realize long-distance sensing, while its low-frequency response is influenced by accumulated phase noise between the signal and local oscillator; direction detection-based DAS has an excellent low-frequency response, while its sensing distance is limited. Thus, coherent detection-based DAS should contribute to reducing accumulated phase noise to realize low-frequency sensing, and direction detection-based DAS should pay attention to suppressing fiber nonlinear effects to extend sensing distance. In addition, the seismic waves propagating in the ocean are weak signals with large background noise. The DAS system should have a sensitivity of at least nε to detect and separate seismic signals [36]. The spatial resolution of DAS is determined by gauge length, and the selection of gauge length will affect the SNR of detection data, which is commonly designed as 10–40 m [57]. 

The comparison of these fiber-optic sensing techniques is shown in Table 1. Both optical interferometry and optical polarimeter can realize seismic monitoring with a long distance through transmitted light. However, it is difficult to accurately locate the earthquake source since the entire cable is used as a single sensor [58]. FBG can reach a high resolution [59], but it is difficult to detect signals less than 1 Hz due to low-frequency fluctuations of the laser. DAS is a fully distributed high-fidelity seismic observation technology with high measurement density and mass observational data; nevertheless, the sensing distance is limited [56,57].

## 4. Conclusions and Outlook

In conclusion, four types of fiber-optics seismic monitoring techniques based on submarine cables optical interferometers, FBG, optical polarimeter, and DAS are discussed in this review. The implementation of fiber-optic sensors enables the realization of long-term, real-time, high-density earthquake detection, which can improve existing observation capabilities in submarine seismic monitoring. The corresponding applications of seafloor earthquake detection are analyzed to summarize the features of each optical fiber sensing technique. The implementation of fiber-optic sensors based on submarine cables has brought great benefits to submarine seismology.

In Table 1, we summarize the key performance parameters of research mentioned in Section 3. For interferometer-based techniques, we find that the sensing principles are various. The variation of strain, phase, frequency, and pressure could be utilized for seismic monitoring, respectively. The sensing length ranges from several hundred meters to thousands of kilometers and the frequency response ranges from 0.004 to 200 Hz. Most techniques have been applied for practical seismic monitoring events. For polarimeter-based techniques, the earthquake information is inverted by the polarization state variation of the transmitting light. Its sensing length can reach up to 10,000 km. For FBG-based techniques, the acceleration of the seismic waves can be obtained through the shift of the wavelength. Seismic information lower than 1 Hz is difficult to detect because its frequency response is limited by frequency fluctuation of the laser. For DAS-based techniques, the variation of the strain of the sensing fiber is utilized for seismic monitoring. The noise floors of these techniques are about 10 nε and the frequency response ranges from 0.001 to 100 Hz, while the sensing length is limited below 100 km.

As an outlook, the future work of each optical seismometer based on submarine cables might be focused on the following. (1) For optical interferometers, a high-density observation array should be constructed to accurately locate the source place; (2) for FBG, a broad detection bandwidth with the low-frequency response should be developed to realize low-frequency natural submarine earthquake detection; (3) for optical polarimeters, the accumulated environmental noise should be suppressed to detect small-sized earthquake events; (4) for DAS, the repeater amplification technology of the submarine cables and the suppression methods of nonlinear effects should be considered to extend its sensing distance.

There is a large amount of telecommunication cables around the world, which can be excellent carriers for fiber-optic sensors to realize submarine seismic monitoring. The key problem is how to utilize these cables to realize earthquake detection without sacrificing the quality of data communication. The establishment of integrated sensing and communication system will become one of the significant directions of future development for fiber-optic sensing technology [60,61,62].

## Figures and Tables

**Figure 1 sensors-23-05600-f001:**
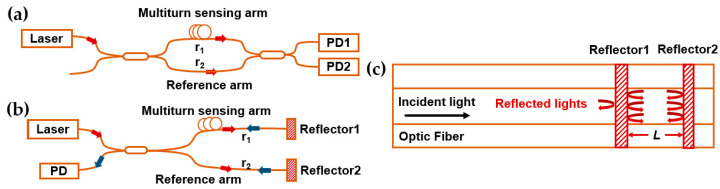
Principles of the (**a**) Mach–Zehnder interferometer, (**b**) Michelson interferometer, (**c**) Fabry–Perot interferometer.

**Figure 2 sensors-23-05600-f002:**
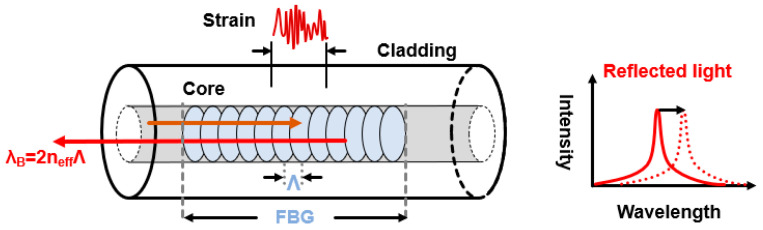
Principle of the FBG sensor.

**Figure 3 sensors-23-05600-f003:**
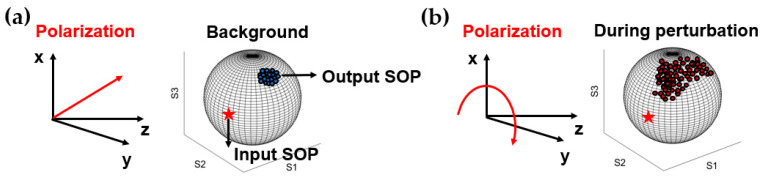
Principle of the polarimeter. (**a**) the SOP before perturbation, (**b**) the SOP under perturbation.

**Figure 4 sensors-23-05600-f004:**
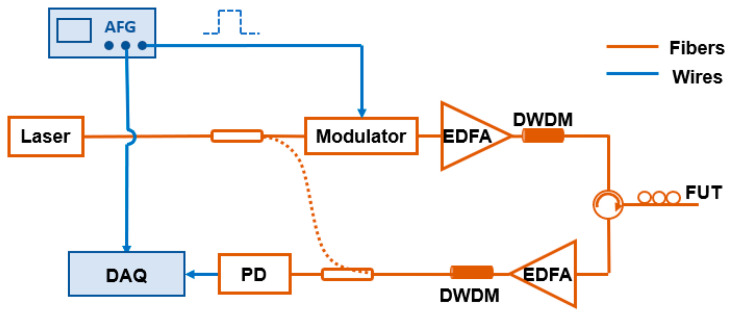
Principle of coherent detection-based DAS (with dashed lines) and direct detection-based DAS (without dashed lines).

**Figure 5 sensors-23-05600-f005:**
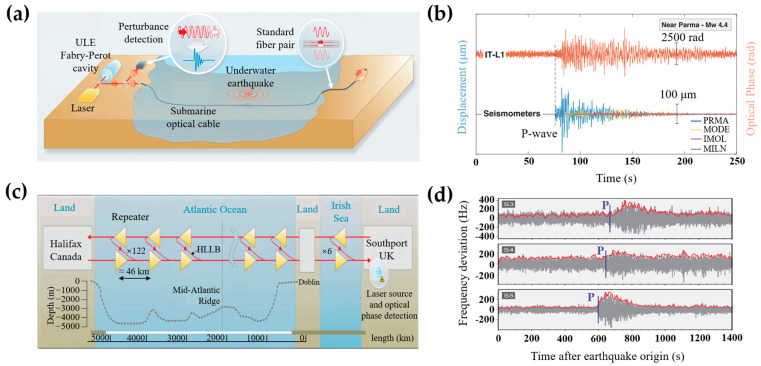
Seismic applications based on the optical interferometer. (**a**) Structure of the USLI system, (**b**) results of earthquake detection, reprinted with permission from [15], (**c**) structure of the USLI observation arrays, (**d**) results of the earthquake location, reprinted with permission from [18].

**Figure 6 sensors-23-05600-f006:**
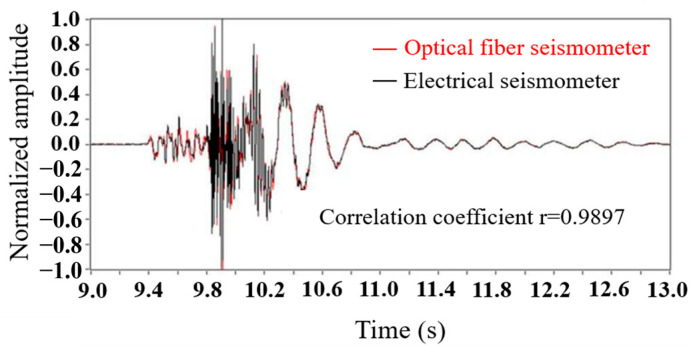
Comparison of detection between optical seismometer and electrical seismometer, reprinted with permission from [37].

**Figure 7 sensors-23-05600-f007:**
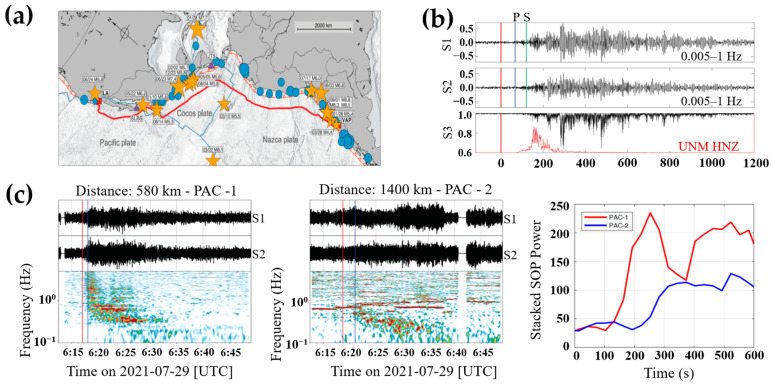
Seismic applications based on optical polarization. (**a**) Experimental submarine cable [16]; (**b**) detection of seismic waves, reprinted with permission from [16]; (**c**) earthquake source location between two submarine cables, reprinted with permission from [23].

**Figure 8 sensors-23-05600-f008:**
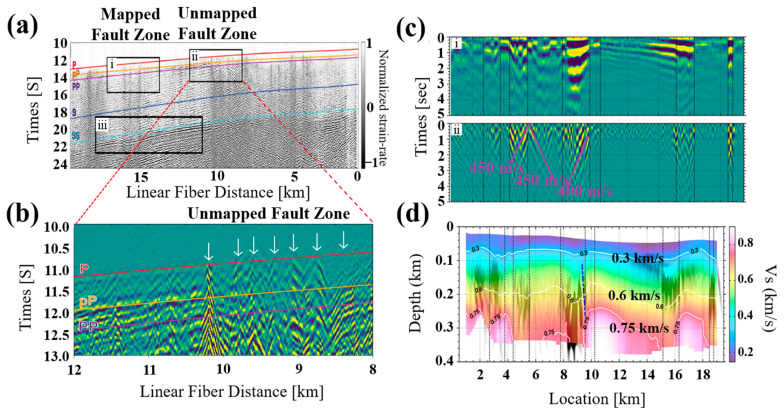
Seismic applications based on coherent detection DAS. (**a**) Seismic observation of DAS, (i) mapped submarine fault locations, (ii) an unmapped fault zone, (iii) wavefront delay in mapped fault zone, (**b**) unmapped submarine fault locations, reprinted with permission from [17], (**c**) autocorrelation image from microseism noise, (i) autocorrelation image from oceanic microseism noise, (ii) the separated scattered scholte waves from autocorrelation profile, (**d**) integrated two-dimension vs. image, reprinted with permission from [45].

**Figure 9 sensors-23-05600-f009:**
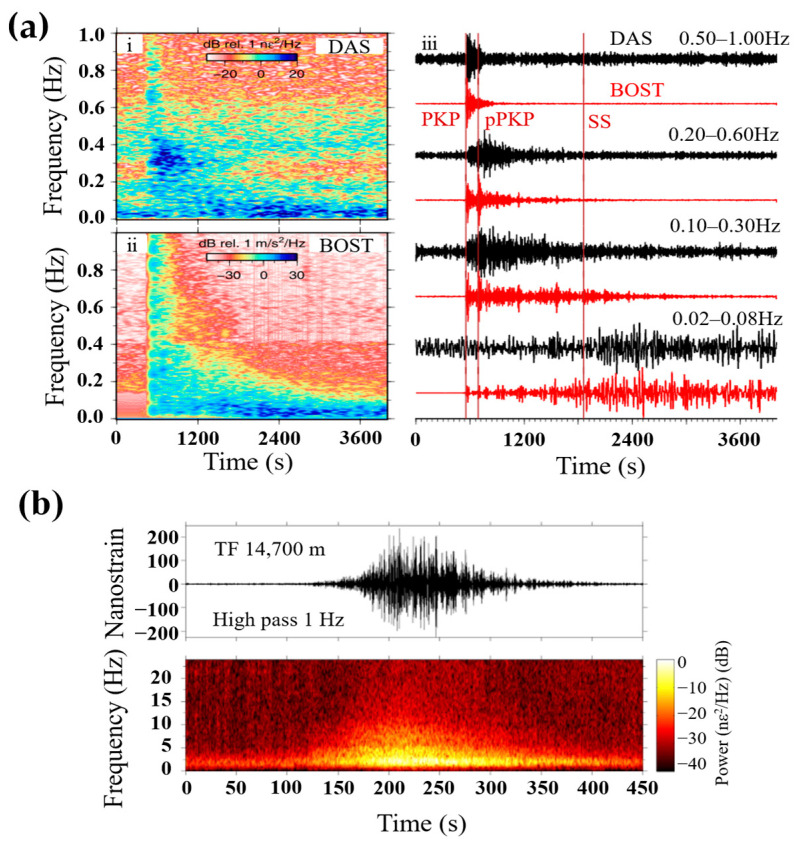
Seismic applications based on direct detection DAS. (**a**) Seismic wave detection, (i) PSD of DAS recordings (ii) PSD of broadband seismometer recordings, (iii) comparison between earthquake signals detected by HDAS and the nearby broadband seismometer, reprinted with permission from [49], (**b**) hydroacoustic T waves detection, reprinted with permission from [50].

**Table 1 sensors-23-05600-t001:** Comparison of various optical seismic monitoring technologies using submarine cables.

Optical Principle	Sensing Length	Noise Floor	Frequency Response	Practical Application
Interferometer	NA [27]	30 ng/√Hz (10 Hz)	0.3–200 Hz	Yes
	200 m [28]	36 nε (RMS variation)	0.004–1.6 Hz	Yes
	NA [30]	6.74 ng/√Hz (1–50 Hz)	0.16–50 Hz	No
	96 km [15]	1 rad/√Hz (1 Hz)	0.01–20 Hz	Yes
	5860 km [18]	1 MHz/√Hz (1 Hz)	0.01–5 Hz	Yes
	125.7 m [32]	100 μPa/√Hz (Average value)	1–80 Hz	Yes
Polarimeter	10500 km [16]	0.03/√Hz (1 Hz)	0.01–10 Hz	Yes
FBG	100 km [33]	0.8 pm/gal	5–50 Hz	Yes
	100 km [34]	NA	3–250 Hz	Yes
	400 m [35]	0.05 pm/gal	10–200 Hz	No
DAS	20 km [17]	1 nε/√Hz (1 Hz)	0.001–10 Hz	Yes
	20 km [45]	1 nε/√Hz (1 Hz)	0.5–10 Hz	Yes
	42 km [49]	10 nε/√Hz (1 Hz)	0.01–10 Hz	Yes
	60 km [50]	100 pε/√Hz (1 Hz)	0.05–24 Hz	Yes
	41.5 km [51]	2 nε/√Hz (1 Hz)	0.2–20 Hz	Yes
	50 km [55]	3 nε/√Hz (1 Hz)	0.01–100 Hz	Yes

## Data Availability

Not applicable.

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
