# Peer review of "Principles and Applications of Seismic Monitoring Based on Submarine Optical Cable"

_sensors, 2023, doi:10.3390/s23125600_

Round 1

Reviewer 1 Report

1. It is understood that φ-OTDR technology also has the potential to be used in underwater acoustic detection, and relevant literature review and citations should be added.

2. It is suggested that the characteristics of seismic wave detection and its special requirements for sensors should be specially supplemented.

Moderate editing of English language required.

Author Response

Response to Reviewer 1 Comments

Thanks for the reviewer’s comments and professional advice, which help us improve the academic rigor of our article. We have made modifications on the manuscript based on the useful suggestions.

Point 1: It is understood that φ-OTDR technology also has the potential to be used in underwater acoustic detection, and relevant literature review and citations should be added.

Response 1: Thanks for the reviewer’s useful suggestion, there are some report about underwater acoustic detection utilized φ-OTDR(DAS), we have added the relevant content in section 3.4., which is marked in red and underlined from line 406 to 415 in the revised manuscript. The references 56-58 are cited as the relevant literature reviews. The details are shown as followings: “Moreover, the DAS system can also serve as hydrophones for earthquake detection [53]. In 2021, Shanghai Institute of Optics and Fine Mechanics, Chinese Academy of Sciences, developed an underwater DAS array to realize the orientation of underwater acoustic signal source [54]. In the same year, Matsumoto et al utilized the submarine DAS system to conduct underwater seismic acoustic waves monitoring around Shikoku Island in Japan. In the air-gun shot field experiments, the hydroacoustic signals record-ed by DAS have a lower frequency response to 0.1 Hz, while the submarine self-noise is higher in the frequency range above 5 Hz compared to the nearby hydrophones [55]. It has been shown that seismic signals induced by a regional earthquake can be identified in the DAS measurement.”

Point 2: It is suggested that the characteristics of seismic wave detection and its special requirements for sensors should be specially supplemented.

Response 2: Thanks for the reviewer’s useful suggestion, the characteristics of seismic wave detection and its special requirements have been discussed before in section 3.4 from line 416 to 434 in the manuscript. The details are shown as followings: “Seismic waves are broadband signals with frequencies ranging from 10-4 to 100 Hz, while frequencies between 0.01 and 10 Hz are typically used in earthquake detection [56]. Excellent low-frequency response is the premise to detect entire earthquake signals. Sever-al seismically active areas are hundreds of kilometers away from the island or mainland. Seismic events occurring in these areas would become weak at the time they reach the seismometers, resulting in distortion of the detected signal. Broad coverage is necessary to detect high-fidelity earthquake signals. Therefore, the DAS systems designed for subma-rine earthquake detection should possess the ability to realize very-low-frequency detec-tion over a long distance. Coherent detection-based DAS is able to realize long-distance sensing, while its low-frequency response is influenced by accumulated phase noise be-tween the signal and local oscillator; direction detection-based DAS has an excellent low-frequency response, while its sensing distance is limited. Thus, coherent detection-based DAS should contribute to reducing accumulated phase noise to realize low-frequency sensing, and direction detection-based DAS should pay attention to suppressing fiber nonlinear effects to extend sensing distance. In addition, the seismic waves propagating in the ocean are weak signals with large background noise. The DAS system should have a sensitivity of at least nε to detect and separate seismic signals [57]. The spatial resolution of DAS is determined by gauge length, and the selection of gauge length will affect the SNR of detection data, which is commonly designed as 10-40 m [58].”

Thank you very much for your attention and time.

Reviewer 2 Report

The authors provides a brief survey on the use of optical fiber technology for seismic monitoring. The review is quite well balanced but some aspects should be improved before publication.

- The description of the principle of operation of optical interferometer based and FBG-based seismic monitoring could be improved. There is one important aspect that has been omitted. In DAS and optical polarimeter-based monitoring system, generally, the optical fiber itself is the sensing cable. The same cannot be said for FBGs and interferometer based monitoring systems. Indeed, the cavity/FBG should be integrated with same transducer (i.e. sensing material) to increase the sensitivity to the acoustic/seismic wave. Eventually the authors could limit the review to configurations using the “optical fiber as sensing cable” but this restriction should be clearly stated (and possibly motivated) at the beginning.

- Table I is inconsistent. Each number in the table should be referenced. Anyways, I suggest to remove this table (at least in this form). A comparison between different technologies would be useful but you should take into account several aspects (not considered in the table):

1) earthquakes can be measured in displacement (seismometers), acceleration (seismic accelerometers ), pressure (hydrophones) but not really in strain. The strain is not a property of the earthquake but it is the indirect effect of the earthquake on the optical fiber. Therefore, it is not suitable in order to compare different technologies. Eventually, the strain resolution is a parameter useful to compare different DAS systems.

2) the final system resolution accounts both for the sensor and the interrogation system. Therefore, each technology could feature different performances according to the different implementation. In this regard, you could provide the best performances for each technology but I think an issue will arise. Is there a single implementation involving the best performance (best resolution, largest bandwidth,.etc).

Probably a comparison could have sense by considering the main implementations for each technology (in different rows)

-I recommend to carefully review the FBG based monitoring systems section (3.2). Reference 38 relies on a fiber laser. In this configuration the FBGs just acts like mirrors. The sensing element is the fiber laser (not the FBG). We can still see it as a FBG based monitoring system but differences among the different FBGs configurations should be better highlighted in the text.

- An interferometric based monitoring system has been recently demonstrated for earthquake detection. “Field demonstration of an optical fiber hydrophone for seismic monitoring at Campi-Flegrei caldera”, Optics & Laser Technology,Volume 158, Part A, 2023, 108920, ISSN 0030-3992, https://doi.org/10.1016/j.optlastec.2022.108920. It should find room in this paper.

- A final comparison among the papers reviewed in section 3 should be provided. The comparison among different technologies should be carried out by using key performances parameters related to the seismic waves detection. It is important to understand which is the noise floor provided (i.e. microPa/sqrt(Hz), ..) by each technology  and in which frequency range.

Other relevant aspects could be underlined in the table. The selection of parameters to be added in the table is a choice of the authors, to support their final discussion (I.e. they could underline, if the practical application of the submarine earthquake detection is demonstrated or not).

Clearly, not all the papers provide the same information but this is not a great issue. Not Avalaible (NA) can be used. (This point is connected to my previous comment to the table I, but whatever is the authors reply to the previous comment, this additional table must be provided after the section 3)

This table should be commented in section “conclusions and outlook”

Some minor aspect:

- DAS acronym should be defined.

- there is a typos in the name of the paragraph 2.2

Author Response

Response to Reviewer 2 Comments

Thanks for the reviewer’s comments and professional advice, which help us improve the academic rigor of our article. We have made modifications on the manuscript based on the useful suggestions.

Point 1: The description of the principle of operation of optical interferometer-based and FBG-based seismic monitoring could be improved. There is one important aspect that has been omitted. In DAS and optical polarimeter-based monitoring system, generally, the optical fiber itself is the sensing cable. The same cannot be said for FBGs and interferometer-based monitoring systems. Indeed, the cavity/FBG should be integrated with same transducer (i.e. sensing material) to increase the sensitivity to the acoustic/seismic wave. Eventually the authors could limit the review to configurations using the “optical fiber as sensing cable” but this restriction should be clearly stated (and possibly motivated) at the beginning.

Response 1: Thanks for the reviewer’s useful suggestion. Whether the optical fiber itself serves as the sensing cables is important to distinguish different optical seismic monitoring techniques. We have added the discussion of these operating principles in section 1, which is marked in red and underlined from line 66 to 71 in the revised manuscript. The details are shown as followings: “For these techniques, the submarine optical cables could not only be utilized as the transmission media but also serve as the sensing element. The optical interferometers and fiber Bragg grating (FBG) mainly adopt submarine cables for data transmission, meanwhile, the optical polarimeter and DAS system utilized submarine cables to perceive external information.”

Point 2: Table I is inconsistent. Each number in the table should be referenced. Anyways, I suggest to remove this table (at least in this form). A comparison between different technologies would be useful but you should take into account several aspects (not considered in the table):

1) earthquakes can be measured in displacement (seismometers), acceleration (seismic accelerometers ), pressure (hydrophones) but not really in strain. The strain is not a property of the earthquake but it is the indirect effect of the earthquake on the optical fiber. Therefore, it is not suitable in order to compare different technologies. Eventually, the strain resolution is a parameter useful to compare different DAS systems.

2) the final system resolution accounts both for the sensor and the interrogation system. Therefore, each technology could feature different performances according to the different implementation. In this regard, you could provide the best performances for each technology but I think an issue will arise. Is there a single implementation involving the best performance (best resolution, largest bandwidth,.etc).

Probably a comparison could have sense by considering the main implementations for each technology (in different rows)

Response 2: Thanks for the reviewer’s useful suggestion. It is inappropriate to only use strain variation to assess the capability of earthquake monitoring, we consider to change the table. The different key performance parameters for each technology would be provided in the additional table in line 451. The performances given in the table are the best performance and could be obtained at the same time.

Point 3: I recommend to carefully review the FBG based monitoring systems section (3.2). Reference 38 relies on a fiber laser. In this configuration the FBGs just acts like mirrors. The sensing element is the fiber laser (not the FBG). We can still see it as a FBG based monitoring system but differences among the different FBGs configurations should be better highlighted in the text.

Response 3: Thanks for the reviewer’s useful suggestion. We have discussed whether the FBGs serve as the sensing element in the mentioned research in section 3.2. The different functions of FBGs in these techniques are highlighted in red and underlined in section 3.2 from line 240 to 252. 

Point 4: An interferometric based monitoring system has been recently demonstrated for earthquake detection. “Field demonstration of an optical fiber hydrophone for seismic monitoring at Campi-Flegrei caldera”, Optics & Laser Technology,Volume 158, Part A, 2023, 108920, ISSN 0030-3992, https://doi.org/10.1016/j.optlastec.2022.108920. It should find room in this paper.

Response 4: Thanks for the reviewer’s useful suggestion. The relevant content in the work “Field demonstration of an optical fiber hydrophone for seismic monitoring at Campi-Flegrei caldera” has been supplemented in section 3.1, which is marked in red and underlined from line 223 to 228. The details are shown as followings: “The University of Sannio developed an MI-based fiber-optic hydrophone for earthquake monitoring in 2023 [32]. The average noise floor of the system is about 100 μPa/√Hz in 1–80 Hz. In the field experiments, several earthquake events were detected and the results were compared with a reference hydrophone sensor. Both hydrophones accurately sensed and recorded the seismic waves and the correlation coefficients between the recorded trials are higher than 85 %. “

Point 5: A final comparison among the papers reviewed in section 3 should be provided. The comparison among different technologies should be carried out by using key performances parameters related to the seismic waves detection. It is important to understand which is the noise floor provided (i.e. microPa/sqrt(Hz), ..) by each technology and in which frequency range.

Other relevant aspects could be underlined in the table. The selection of parameters to be added in the table is a choice of the authors, to support their final discussion (I.e. they could underline, if the practical application of the submarine earthquake detection is demonstrated or not).

Clearly, not all the papers provide the same information but this is not a great issue. Not Avalaible (NA) can be used. (This point is connected to my previous comment to the table I, but whatever is the authors reply to the previous comment, this additional table must be provided after the section

This table should be commented in section “conclusions and outlook”

Response 5: Thanks for the reviewer’s useful suggestion. The comparison among different technologies mentioned in section 3 has been provided in the additional table in the final part of section 3. The title of the additional table is marked in red and underlined in line 447. The noise floor, frequency response range, and sensing length, as the key performance parameters have been given in this additional table. The details are shown as followings:

Table 1. Comparison of various optical seismic monitoring technologies using submarine cables

Optical Principle

Sensing Length

Noise Floor

Frequency Response

Practical Application

Interferometer

NA [27]

30 ng/√Hz

(10 Hz)

0.3 – 200 Hz

Yes

200 m [28]

36 nε

(RMS variation)

0.004 – 1.6 Hz

Yes

NA [30]

6.74 ng/√Hz

(1 – 50 Hz)

0.16 – 50 Hz

No

96 km [15]

1 rad/√Hz

(1 Hz)

0.01 – 20 Hz

Yes

5860 km [18]

1 MHz/√Hz

(1 Hz)  

0.01 – 5 Hz

Yes

125.7 m [32]

100 μPa/√Hz

 (Average value)

1 – 80 Hz

Yes

Polarimeter

10500 km [16]

0.03 /√Hz

(1 Hz)

0.01– 10 Hz

Yes

FBG

100 km [33]

0.8 pm/gal

5 – 50 Hz

Yes

100 km [34]

NA

3 – 250 Hz

Yes

400 m [35]

0.05 pm/gal

10 – 200 Hz

No

DAS

20 km [17]

1 nε/√Hz

(1 Hz)

0.001 – 10 Hz

Yes

20 km [45]

1 nε/√Hz

(1 Hz)

0.5 – 10 Hz

Yes

42 km [49]

10 nε/√Hz

(1 Hz)

0.01 – 10 Hz

Yes

60 km [50]

100 pε/√Hz

(1 Hz)

0.05 – 24 Hz

Yes

41.5 km [51]

2 nε/√Hz

(1 Hz)

0.2 – 20 Hz

Yes

50 km [55]

3 nε/√Hz

(1 Hz)

0.01 – 100 Hz

Yes

The comment on the additional table is marked in red and underlined in section “conclusions and outlook” from line 457 to 471. The details are shown as followings: “In table I, we summarize the key performance parameters of research mentioned in section 3. For interferometers-based techniques, we find that the sensing principles are various. The variation of strain, phase, frequency, and pressure could be utilized for seismic monitoring, respectively. The sensing length ranges from several hundred meters to thousands of kilometers and the frequency response ranges from 0.004 to 200 Hz. Almost techniques have been applied for practical seismic monitoring events. For Polarimeter-based technique, the earthquake information is inverted by the polarization state variation of the transmitting light. Its sensing length can reach up to 10000 km. For FBG-based techniques, the acceleration of the seismic waves can be obtained through the shift of the wavelength. The seismic information lower than 1 Hz is difficult to detect for its frequency response is limited by frequency fluctuation of the laser. For DAS-based techniques, the variation of the strain of the sensing fiber is utilized for seismic monitoring. The noise floors of these techniques are about 10 nε and the frequency response ranges from 0.001 to 100 Hz, while the sensing length is limited below 100 km.”

Point 6: DAS acronym should be defined; there is a typos in the name of the paragraph 2.2.

Response 6: Thanks for the reviewer’s useful suggestion. The acronym of DAS have been defined before in line 66 in the manuscript. The typos in the name of the paragraph 2.2 in line 112 has been corrected, which is marked in red and underlined.

Thank you very much for your attention and time.

Round 2

Reviewer 2 Report

The authors clarified the potentially misleading aspects which I underlined in my previous report

The added final table is much more informative and support the final discussion 

In my opinion the paper can be accepted for publication